# Understanding the Mechanism of Light-Induced Age-Related Decrease in Melanin Concentration in Retinal Pigment Epithelium Cells

**DOI:** 10.3390/ijms241713099

**Published:** 2023-08-23

**Authors:** Alexander E. Dontsov, Marina A. Yakovleva, Alexander A. Vasin, Alexander A. Gulin, Arseny V. Aybush, Viktor A. Nadtochenko, Mikhail A. Ostrovsky

**Affiliations:** 1Emanuel Institute of Biochemical Physics of Russian Academy of Sciences, 119334 Moscow, Russia; lina.invers@gmail.com (M.A.Y.); ostrovsky3535@mail.ru (M.A.O.); 2Federal Research Center of Chemical Physics, Russian Academy of Sciences, N.N. Semenov RAS, 119991 Moscow, Russia; a2vasin@yandex.ru (A.A.V.); aleksandr.gulin@phystech.edu (A.A.G.); aiboosh@gmail.com (A.V.A.); ihf-ran@yandex.ru (V.A.N.)

**Keywords:** human eye, retinal pigment epithelium, melanosomes, melanolipofuscin granules, visible light, superoxide, melanin degradation

## Abstract

It is known that during the process of aging, there is a significant decrease in the number of melanosomes in the retinal pigment epithelium (RPE) cells in the human eye. Melanosomes act as screening pigments in RPE cells and are fundamentally important for protection against the free radicals generated by light. A loss or change in the quality of melanin in melanosomes can lead to the development of senile pathologies and aggravation in the development of various retinal diseases. We have previously shown that the interaction between melanin melanosomes and superoxide radicals results in oxidative degradation with the formation of water-soluble fluorescent products. In the present study, we show, using fluorescence analysis, HPLC, and mass spectrometry, that visible light irradiation on melanolipofuscin granules isolated from RPE cells in the human eye results in the formation of water-soluble fluorescent products from oxidative degradation of melanin, which was in contrast to lipofuscin granules and melanosomes irradiation. The formation of these products occurs as a result of the oxidative degradation of melanin by superoxide radicals, which are generated by the lipofuscin part of the melanolipofuscin granule. We identified these products both in the composition of melanolipofuscin granules irradiated with visible light and in the composition of melanosomes that were not irradiated but were, instead, oxidized by superoxide radicals. In the melanolipofuscin granules irradiated by visible light, ions that could be associated with melanin oxidative degradation products were identified by applying the principal component analysis of the time-of-flight secondary ion mass spectrometry (ToF-SIMS) data. Degradation of the intact melanosomes by visible light is also possible; however, this requires significantly higher irradiation intensities than for melanolipofuscin granules. It is concluded that the decrease in the concentration of melanin in RPE cells in the human eye with age is due to its oxidative degradation by reactive oxygen species generated by lipofuscin, as part of the melanolipofuscin granules, under the action of light.

## 1. Introduction

The retinal pigment epithelium (RPE) cells in the eye contain three types of pigment granules: melanosomes, lipofuscin granules, and melanolipofuscin granules. Melanosomes are specialized organelles in RPE cells that contain a protein part and a eumelanin-type polymer. All melanosomes in RPE cells are formed right at the time of embryonic development and perform their functions throughout the life of an individual. Conversely, lipofuscin and melanolipofuscin granules are formed much later and accumulate in the RPE cells as a function of age. Changes in the density of melanosomes and lipofuscin granules in the RPE cells are considered to be hallmarks of aging or the development of various retinal diseases, including Stargardt disease and age-related macular degeneration [1]. Melanosomes perform the function of screening photoreceptor cells from excessive illumination and protecting the cells against antioxidants from free-radical oxidation caused by irradiation and reactive oxygen species [2,3,4,5,6]. With age, there is a significant decrease in the number of melanosomes in RPE cells. Whereas melanosomes occupy up to 8% of the RPE cell volume before the age of 20 years, this volume gradually decreases to 3.5% between the ages of 40 and 90 years [7,8]. At the same time, the total concentration of melanin in RPE cells decreases by 2.5 times [8]. Simultaneously with age, the density of lipofuscin and melanolipofuscin granules increases in RPE cells [7,9]. A decrease in the density of the melanosomes with age may be due to the fact that they conjugate with lipofuscin granules [10,11,12,13] or with partially degraded phagosomes [14,15,16,17]. This results in the formation of mixed melanolipofuscin granules which, along with lipofuscin granules, can occupy up to a third of the cytoplasmic space in RPE cells, in people over 70 years old [7]. However, this process alone cannot lead to a decrease in the total concentration of melanin in RPE cells.

Melanin content has been suggested to decrease due to its photooxidative and/or oxidative degradation [8,18]. Oxidative degradation of RPE cellular melanosomes leads, on the one hand, to a decrease in their antioxidant activity, and, on the other hand, to the appearance of pro-oxidant properties and an increase in photoreactivity [5,19,20,21,22].

However, the specific mechanisms of melanosome biodegradation in RPE cells are still not known. The most probable is the oxidative degradation of melanosomes as a result of their interaction with reactive oxygen species (ROS). ROS, such as superoxide radicals and/or hydrogen peroxide, are known to cause the degradation of the melanin polymer [18,23,24,25,26] and the formation of water-soluble fluorescent products [18,27,28,29,30]. In RPE cells in the eye, reactive oxygen species can arise from both normal metabolic processes and photoinduced oxidation. A well-known light-induced generator of reactive oxygen species in RPE cells is lipofuscin bisretinoids, which include N-retinylidene-retinyl ethanolamine (A2E). When irradiated by visible light, especially in the blue range, lipofuscin granules from RPE cells reduce oxygen to superoxide radicals [31,32,33,34]. At the same time, the intensity of superoxide generation by lipofuscin should increase with age since the number of lipofuscin granules increases sharply with aging.

We have previously shown that upon irradiation with blue light (450 nm; 4 mW/cm^2^) of RPE melanosomes, fluorescent melanin degradation products are virtually not formed, whereas irradiation of a mixture of melanosomes and lipofuscin granules leads to the appearance of melanin degradation products [18]. On this basis, it was suggested [18] that age-related destruction of melanin by superoxide radicals occurs under the action of light in melanolipofuscin granules that contain a source of superoxide radicals, lipofuscin. In this case, the age-related decrease in the concentration of melanin in RPE cells would result from a decrease in the number of melanosomes due to their fusion with lipofuscin granules, and the subsequent degradation of the melanin that was already inside these complex granules. Therefore, due to the degradation of melanin, the specific gravity of melanolipofuscin granules will decrease, which will lead to the accumulation of mixed granules with small admixtures of melanin that had not yet been destroyed by the superoxide radicals. This process will lead to a general decrease in the melanin concentration in RPE cells.

In the present study, we showed that blue light irradiation of melanolipofuscin granules in RPE cells in the human eye, in contrast to irradiation of lipofuscin granules and melanosomes, resulted in the accumulation of products from the oxidative degradation of melanin caused by superoxide radicals. It has also been shown that the products of the oxidative degradation of RPE melanosomes are photoinducible generators of superoxide and contain carbonyl compounds. Therefore, we proposed a scheme of oxidative degradation of melanin and lipofuscin, which were inside the melanolipofuscin granules, under the action of light.

## 2. Results and Discussion 

### 2.1. Spectral Fluorescence Studies of Melanin Oxidative Degradation Products

Incubation of a suspension of human RPE melanosomes with relatively low concentrations of potassium superoxide results in the accumulation of water-soluble products, with relatively high absorption values in the short-wavelength region of the spectrum (less than 400 nm; Figure 1A, curve 2). Water-soluble products obtained from intact, non-oxidized melanosomes do not absorb in the long-wavelength region and absorb poorly in the short-wavelength region of the spectrum (Figure 1A, curve 1). This is probably due to the process of melanin oxidation by superoxide radicals, which leads to the accumulation of water-soluble, low-molecular products from its destruction in the incubation medium. Indeed, as can be seen from Figure 1B, melanosome oxidation by superoxide radicals results in products with emission maxima of 460–480 nm (curve 2; excitation at 365 nm) and 530 nm (curve 4; excitation at 470 nm). At the same time, the water-soluble fractions obtained from the initial, non-oxidized suspension of melanosomes showed almost no fluorescent properties (Figure 1B, curves 1 and 3, respectively).

Incubation of a suspension of melanolipofuscin granules with potassium superoxide also leads to the appearance of degradation products from fluorescent melanin, though to a lesser extent, which is apparently due to the lower melanin content in these granules, compared to melanosomes (Figure 2A,B, curves 1 for water-soluble products melanosome destruction and curves 2 for water-soluble destruction products of melanolipofuscin granules).

### 2.2. Characterization and Potential Phototoxicity of Melanin Oxidative Degradation Products from RPE Pigment Granules

The next stage of the work was to study the nature and physicochemical properties of the water-soluble products formed during the oxidative degradation of RPE melanosomes, induced by potassium superoxide. Potassium superoxide at millimolar concentrations disrupts RPE melanosomes through the formation of fluorescent products with an emission maximum of 520–525 nm. Figure 3A shows the kinetics of the accumulation of melanin degradation products by melanosomes. The degradation products formed during the oxidative degradation of melanin contain carbonyl compounds. This is evidenced by the data obtained using time-of-flight secondary ion mass spectrometry. As can be seen from the diagram (Figure 3C), under the action of potassium superoxide on melanosomes, there is a significant increase in carbonyl ions (*m*/*z* = 29 − CHO^+^ ion, *m*/*z* = 60 − C_2_H_4_O_2_^+^ ion, *m*/*z* = 69 − C_4_H_7_O^+^ ion). Ions with *m*/*z* = 60 demonstrate the greatest increase in concentration (an increase of about 1.5 times). The obtained results indicate the presence of aldehydes in the products formed during the oxidation of melanosomes by superoxide radicals. These results are confirmed by experiments on the content of carbonyl compounds that react with thiobarbituric acid (TBA-reactive products) (Figure 3C). As can be seen, the water-soluble fractions obtained from the oxidized samples (Figure 3B, columns 2 and 3) contain significantly more TBA-reactive products than the supernatants of the control and non-oxidized samples (Figure 3B, column 1), while the higher the dose of KO_2_ formed more TBA-reactive products (Figure 3B, columns 2 and 3).

The products from oxidative degradation of melanin in RPE melanosomes exhibited the ability for the light-induced generation of superoxide radicals (Figure 3D). When irradiated with blue light (450 nm), they catalyzed the reduction of cytochrome *c* (transition of Fe^3+^ to Fe^2+^ form) and nitro blue tetrazolium to formazan (not shown in the figures). The process is inhibited in the presence of superoxide dismutase. 

The obtained results indicate that the oxidation of melanosomes with potassium superoxide results in the formation of carbonyl products that are readily soluble in the aqueous phase. The degree of melanin degradation depends on the melanin/superoxide ratio. Increasing the superoxide concentration leads to increased accumulation of TBA-reactive products (Figure 3B). Calculations show that one RPE cell, with a volume of 2 × 10^−9^ mL and containing about 10–20 picograms of melanin, can accumulate up to 0.3 nmol of TBA-reactive products when completely destroyed. Aldehydes and ketones contained in the water-soluble fraction, obtained as a result of the oxidative degradation of melanosomes, were also found to be quite stable owing to the content of TBA-reactive products in them, which remained virtually unchanged (not shown in the figure). The appearance of such long-lived and active chemicals during the oxidative degradation of RPE cells by melanosomes may contribute to the development of various eye pathologies. The revealed ability of the products from the oxidative degradation of melanosomes to the light-dependent generation of superoxide radicals indicates their potential phototoxicity.

### 2.3. Spectral Fluorescence Studies of Melanin Photo-Oxidative Degradation Products

We hypothesize that when melanolipofuscin granules are irradiated with blue light, the melanin contained in them can be destroyed by the action of superoxide radicals generated by lipofuscin bisretinoids during photoexcitation. We assume that blue light irradiation of melanosomes would not lead to a noticeable accumulation of melanin degradation products due to its high resistance to irradiation, and the absence of lipofuscin fluorophores and other photosensitizers in melanosomes. Experiments on the comparative irradiation of RPE melanosomes and melanolipofuscin granules by blue light showed that only upon irradiation of melanolipofuscin granules can a significant excess of fluorescence intensity of water-soluble products be observed, compared to unirradiated granules (Figure 4, curves 2). This was observed at all excitation wavelengths, although was especially pronounced following excitation at 365 nm (Figure 4C). 

The average values of emission maxima for irradiated samples were determined by the results of three assays for wavelength characteristics of water-soluble products of oxidative degradation of melanin, namely 535, 525, and 460 nm. For the products formed following the irradiation of melanolipofuscin granules, the average emission values were +0.15, +1.2, and +16.0, respectively. For melanosomes, these values were negative, except for the emission maximum at 460 nm (+4.0). This indicates the appearance of melanin degradation products upon irradiation of melanolipofuscin granules. In contrast to melanolipofuscin granules, irradiation of melanosomes under the same conditions led to a decrease in the fluorescence intensity by the resulting products, meaning that the differential spectra were in the negative region (Figure 4A,B, curves 1), except for the excitation wavelength of 365 nm (Figure 4C, curves 1). At this excitation wavelength, melanin degradation products were observed, which may be associated with either the presence of a slight admixture of lipofuscin in melanosomes or with an admixture of pro-oxidant degradation products from melanin itself (see below). 

However, an increase in the irradiation intensity of the RPE melanosomes with visible light led to an increase in melanin degradation (Figure 5). The appearance of melanosome degradation products upon their irradiation with high-intensity light can be clearly seen in the chromatogram (Figure 5C). 

### 2.4. Detection of Products from Oxidative Degradation of Melanin in Melanolipofuscin Granules

#### 2.4.1. HPLC Analysis

To detect melanin degradation products in the water-soluble fraction of melanolipofuscin granules irradiated with visible light, we analyzed the chromatograms of the supernatants for all three types of RPE pigment granules: Melanosomes, melanolipofuscin granules, and lipofuscin granules. It was important to show that irradiation of melanolipofuscin granules with visible light formed products that were characteristic of the oxidative degradation of melanin under the action of superoxide radicals. For this purpose, chromatography experiments were carried out on the products formed during the oxidative degradation of RPE melanosomes by potassium superoxide. These data were compared to the chromatograms of the water-soluble products from the light-induced degradation of RPE melanolipofuscin and lipofuscin granules. Figure 6 shows the typical chromatograms of superoxide-oxidized supernatants from melanosomes (A), as well as from melanolipofuscin granules (B), and lipofuscin granules (C), following irradiation with blue LED light.

Figure 6C shows that the water-soluble products formed upon the irradiation of RPE lipofuscin granules are characterized by peaks designated by numbers 6, 9, 10, 12, 13, 15, 16, 18, 20, and 22. On the contrary, the water-soluble degradation products from RPE melanosomes, formed during the oxidation of the granules by superoxide, are characterized by the peaks designated by numbers 2, 3, 4, 5, 7, 8, 11, 14, 17, 19, and 21 (Figure 6A). The standards for the specific melanin degradation products, pyrrole-2,2-dicarboxylic acid (PDCA, peak № 17) and pyrrole-2,3,5-tricarboxylic acid (PTCA, peak № 21), are released at 11.5 min and at 22.1 min, respectively, which is in good agreement with the literature data [35]. 

It is important to note that among the degradation products from RPE melanosomes, we observed peaks for both PDCA and PTCA, which are considered to be the main products from the oxidative degradation of melanin and characterize its initial structure, as well as other compounds that are formed during the degradation of melanosomes following irradiation by intense visible light (Figure 5C). This indicates the existence of two different groups of melanin degradation: the first group is PDCA and PTCA, and the second group is melanin degradation fragments, which exhibit longer wavelength fluorescence.

The water-soluble fraction obtained from irradiated melanolipofuscin RPE granules shows the presence of peaks, which are characteristic of degradation products for both melanosomes and lipofuscin granules (Figure 6B). Thus, in contrast to lipofuscin RPE granules, the water-soluble products from the light-induced degradation of melanolipofuscin RPE granules, irradiated with blue light from an LED lamp (450 nm, 4 mW/cm^2^), contain products characteristic of the oxidative degradation of melanin caused by superoxide radicals (Figure 6B).

The results confirm the assumption that bisretinoid fluorophores present in the melanolipofuscin granules, which generate superoxide radicals under the action of blue light, cause the oxidative degradation of the melanin contained in these granules.

#### 2.4.2. Mass Spectrometry Analysis

The compounds formed as a result of light-induced degradation of RPE pigment granules were characterized using principal component analysis (PCA) for mass spectra obtained by the ToF–SIMS. The mass spectra of samples from melanosomes, melanolipofuscin, and lipofuscin granules were obtained before and after their irradiation by visible light. The peaks that related to the melanin degradation products were identified both in the composition of the melanosomes and in the composition of the melanolipofuscin granules. PCA is routinely applied for ToF–SIMS data to distinguish the differences in sample compositions [36]. The PCA score plot revealed a significant difference between the melanolipofuscin and lipofuscin granule samples (Figure 7).

Principal component 1 (PC1) was mainly related to the difference between melanolipofuscin and lipofuscin, whereas PC2 could be attributed to the difference between the non-irradiated and irradiated states of the RPE granules. PC loadings analysis was applied to make a list of photodegradation fragments that enlarged the ion yield in the melanolipofuscin granules. 

However, the source of the photodegradation fragments remains unclear. It could be either a supernatant or melanin polymer structure. In order to understand the source of the photodegradation products, a suspension of melanosomes containing melanin degradation products was subjected to centrifugation. Two samples were acquired: a precipitate containing melanosomes with an intact melanin polymer structure, and a supernatant, where the small molecules from the melanin degradation products were present in water solution. Mass spectra were analyzed by PCA in the same manner. As a result, the fragments in the supernatants corresponding to the products from the melanin degradation from oxidized melanosomes were revealed and compared to the ions found in the melanolipofuscin. Table 1 shows the masses responsible for the photodegradation products by melanin in both the melanosomes and melanolipofuscin granules. The assignments were proposed according to the exact mass value (Table 1).

The obtained results indicate the presence of melanin degradation products in the water-soluble fraction of melanolipofuscin granules irradiated with blue light. Previously, we found that the concentration of melanin in human RPE melanosomes does not change with age [37]. We can assume that the decrease in melanin concentration with age is not due to its destruction in RPE melanosomes but is caused by its degradation by superoxide radicals generated by lipofuscin bisretinoids, localized in melanolipofuscin granules. In the present study, we have shown that it is superoxide radicals that cause melanin degradation in melanolipofuscin granules, and the resulting products, in turn, are photosensitive generators of superoxide. Considering these facts, a scheme of the mechanism of melanin and lipofuscin degradation in melanolipofuscin granules is proposed to explain the reasons for the decrease in melanin concentration in RPE cells with age (Figure 8).

Light in the presence of oxygen activates the generation of superoxide mediated by lipofuscin fluorophores (Figure 8). The superoxide reacts with the melanin in the melanolipofuscin granule, causing its degradation and the formation of photosensitive degradation products. The latter can partially leave the granule and partially remain in it, which in turn causes the additional formation of superoxide under the action of light. This leads to further degradation of melanin and oxidation of lipofuscin. Carbonyls, which are the toxic products from lipofuscin oxidation, can exit the granule into the cell and damage cytoplasmic proteins in the dark.

## 3. Materials and Methods 

### 3.1. Preparing Products of Oxidative Degradation of RPE Pigment Granules during Oxidation by Superoxide Radicals

Pigment granules were isolated from RPE cells in human cadaveric eyes. Human cadaveric eyes were obtained from the Eye Tissue Bank of the S.N. Fedorov Institute of Eye Microsurgery of the Ministry of Health of Russia from donors without ophthalmic pathologies, aged 40 to 75 years. A total of 40 donor eyes were used. Pigment granules were isolated from retinal pigment epithelium cells, according to the method described in [38]. RPE cells in 0.1 M K–phosphate buffer, pH 7.4, were cooled to 4 °C, and sonicated twice for 40 s at maximum resonance, followed by removal of undamaged cell membranes by centrifugation at 100× *g* for 15 min. The resulting supernatant was centrifuged at 6000× *g* for 15 min, the precipitate was suspended in 0.3 M sucrose and centrifuged in an ultracentrifuge in a sucrose density gradient (2.0; 1.8; 1.6; 1.5; 1.4; 1.2; 1.0; 0.3 M) at 103,000× *g* for 1 h. Then, pigment granules, separated into fractions, were selected and washed with 0.1 M K–phosphate buffer to remove the sucrose. Melanosomes formed a dark brown precipitate, lipofuscin granules formed a layer in the region of a sucrose density of 1.2, and melanolipofuscin granules were localized in the region of 1.6–1.8. The granules were suspended in phosphate buffer and stored in a freezer at −18 °C. Granule concentration counting was carried out in a Goryaev chamber using a “LOMO MikMed-2” fluorescent microscope, equipped with a camera.

Oxidative degradation of granules was carried out by incubating a suspension of granules in 0.1 M K–phosphate buffer, pH 7.4, (1.0 × 10^8^ – 5.0 × 10^8^ granules/mL) with dry potassium superoxide (3–8 mg) at room temperature for 1–2 h. At the end of the reaction, the water-soluble products were separated by centrifugation using a Beckman Allegra 64R centrifuge at 12,000× *g* for 20 min.

Photoinduced degradation of RPE pigment granules was carried out by irradiating them with blue light from an LED source, at an irradiation energy of 4 mW/cm^2^, for 1–2 h. In some experiments, the RPE melanosome photodegradation products were obtained by irradiating them with visible light from a halogen lamp, at an energy of 100 mW/cm^2^, for at least two hours. At the end of the reaction, the water-soluble products were separated by centrifugation using a Beckman Allegra 64R centrifuge at 12,000× *g* for 20 min.

### 3.2. Investigation of the Spectral Characteristics of Degradation Products of Pigment Granules

The degradation products of melanin-containing pigment granules were recorded fluorometrically by measuring the emission intensity at wavelengths of 365 nm, 450 nm, and 470 nm, which corresponded to the maxima of the excitation spectrum of melanin degradation products, and at emission maxima of 460 nm, 525 nm, and 535 nm. In addition, the emission intensities of the melanin degradation products were recorded at an excitation wavelength of 488 nm, which was used in the ophthalmological practice for the non-invasive study of fundus autofluorescence.

The emission intensity was measured on a Shimadzu RF5301PC Spectro fluorophotometer (Kyoto, Japan). To estimate the concentrations of the melanin degradation products, we used a calibration curve, of the dependence of the emission intensity at 520–525 nm, on the concentration of synthetic DOPA melanin completely oxidized by KO_2_. The control samples were incubated for the same time in the dark. When the reaction was completed, the water-soluble degradation products for the granules were obtained by centrifugation at 15,000× *g* for 20 min. Differential fluorescence spectra (light–dark) were plotted to analyze the obtained products.

### 3.3. Measurement of Photoinduced Superoxide Radicals Generated by Water-Soluble Products from Oxidative Destruction of Melanosomes and Melanolipofuscin Granules

The process of the light-induced generation of superoxide radicals by water-soluble products following the oxidative destruction of melanin was estimated by measuring the kinetics for the reduction of cytochrome *c* (Fe^3+^) when superoxide dismutase was inhibited, using a modified method [39]. The reaction mixture contained 0.1 M K–phosphate buffer, pH 7.6, 100 μM cytochrome *c*, 0.05% cetyl trimethyl ammonium bromide, and various amounts of water-soluble melanin oxidative degradation products. The mixture was irradiated by blue light from an LED source at an irradiation energy of 4 mW/cm^2^ and an irradiation wavelength of 450 nm, with constant stirring, after which the absorption maximum was measured at a wavelength of 550 nm (molar absorption coefficient 21 mM^−1^cm^−1^) [40]. Samples were incubated in the dark and samples containing no melanin degradation products were used as controls. The data are expressed as the mean ± SD. The data are presented based on the results of three independent measurements.

### 3.4. Measurement of Products That React with Thiobarbituric Acid (TBA-Reactive Products)

Water-soluble degradation products of melanosomes and melanolipofuscin RPE granules were evaluated for the content of reactive carbonyls reacting with thiobarbituric acid [41]. The concentration of TBA-reactive products was determined spectrophotometrically at a wavelength of 532 nm [42] using a Shimadzu UV-1700 spectrophotometer (Kyoto, Japan). Water-soluble products obtained from native, non-oxidized pigment granules served as controls. The data are expressed as the mean ± SD. The data are presented based on the results of four independent measurements.

### 3.5. Mass Spectrometry of Pigment Granule Destruction Products by ToF–SIMS

Mass spectra were obtained on a TOF.SIMS.5 time-of-flight secondary ion mass spectrometer (ION-TOF, Münster, Germany), with a bismuth cluster beam, which is regularly used in biological studies [43]. The 300 × 300 μm (64 × 64 pixels) analysis area was probed by primary Bi_3_^+^ ions at an energy of 30 keV and with a primary ion dose density of ~4 × 10^11^ ions/cm^2^ in each measurement. At least 10 measurements were recorded for each sample, for both positive and negative ion modes. During the analysis, the electron gun was activated. Processing and identification of the ions were carried out using SurfaceLab 7 software. 

### 3.6. Chromatographic Analysis of Melanin Degradation Products

Chromatographic separation of melanin degradation products in supernatants from melanosomes, melanolipofuscin, and lipofuscin granules in the human eye was carried out on a Knauer chromatograph (Berlin, Germany), with a Diasfer 120 C18 column (4 × 250 mm, sorbent size 5 μm), according to the modernized method described in the work [37]. PDCA and PTCA were kindly provided by Prof. Dr. Kazumasa Wakamatsu (Institute for Melanin Chemistry, Fujita Health University, Toyoake, Japan). A mixture of a 1% solution of formic acid in water (pH, 2.8) and methanol in a ratio of 97:3, by volume, was used as an eluent [37]. The eluent flow rate was 0.8 mL/min. The chromatographic separation products were measured using a Knauer K-2501 photometric detector and a fluorometric detector (RF-10A-xl, Shimadzu, Kyoto, Japan). Detection was performed at 270 nm by absorption, as well as by fluorescence at different excitation and detection wavelengths. Precision for each sample was determined from three independently measured chromatograms for each individual sample.

## 4. Conclusions

The obtained results indicate that water-soluble carbonyls, aldehydes, and ketones, are formed during the oxidative degradation of RPE melanosomes by superoxide radicals, and all exhibit TBA activity. It was found for the first time that water-soluble melanin degradation products are photosensitive and generate superoxide radicals under irradiation by blue light. Consequently, the toxicity of melanin oxidative degradation products on cells can be due to both the content of reactive aldehydes and the ability to generate light-dependent superoxide radicals. Since superoxide radicals cause the oxidation and degradation of melanin, as well as the oxidation and degradation of lipofuscin fluorophores, this can lead to increased light-induced degradation of both lipofuscin and melanin in the melanolipofuscin granules (Figure 8). This fact may also explain the important role of melanin in the process of lipofuscin degradation in RPE cells [19].

Melanin degradation products, which formed as a result of its reaction with superoxide radicals, were identified and characterized using HPLC and mass spectrometry. The same products were found in samples of melanolipofuscin granules irradiated with blue light. This indicates that melanin is degraded by superoxide radicals in these granules. Superoxide radicals in melanolipofuscin granules can be generated during the irradiation of granules by both bisretinoids, which are part of the lipofuscin part of the granule, and photosensitive degradation products from the melanin part of the granule.

Thus, it can be concluded that when melanolipofuscin granules are irradiated by blue light (450 nm, 4 mW/cm^2^), the melanin included in their composition will be destroyed as a result of its reaction with the superoxide radicals. This process probably leads to an age-related decrease in the concentration of melanin in RPE cells in the human eye.

## Figures and Tables

**Figure 1 ijms-24-13099-f001:**
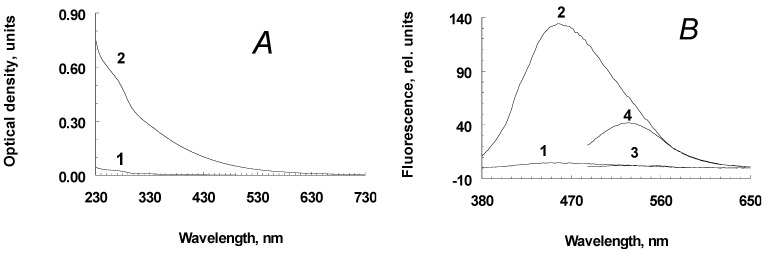
Spectral characteristics of water-soluble destruction products of human RPE melanosomes. (**A**)—Optical absorption (curves 1 and 2—water-soluble products from non-oxidized and superoxide-oxidized melanosomes, respectively); (**B**)—fluorescence spectrum upon excitation at 365 nm (1, 2) and 470 nm (3, 4) (curves 1, 3 and 2, 4—water-soluble products from non-oxidized and superoxide-oxidized melanosomes, respectively). Rel. units—relative units.

**Figure 2 ijms-24-13099-f002:**
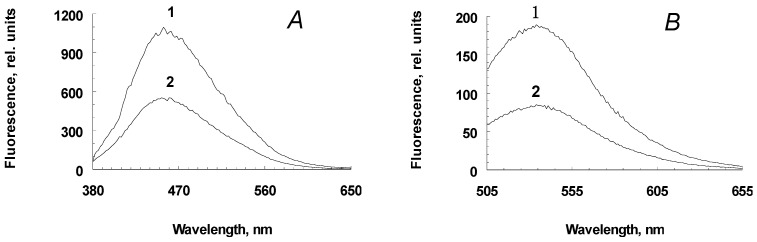
Differential fluorescence spectra (emission of oxidized samples minus emission of original non-oxidized samples) for water-soluble fractions from RPE pigment granules. RPE granules were oxidized with potassium superoxide, the fluorescence excitation wavelengths were 365 nm (**A**) and 488 nm (**B**). Rel. units—relative units. 1—melanosomes, 2—melanolipofuscin granules.

**Figure 3 ijms-24-13099-f003:**
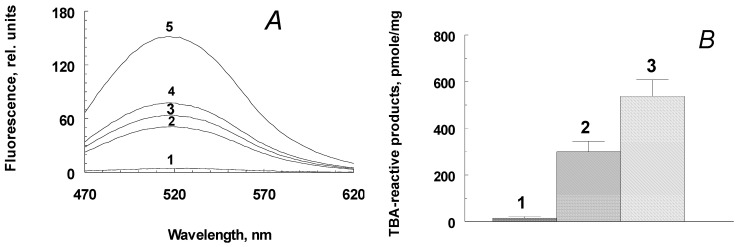
Physicochemical characteristics of the products formed during the interaction between RPE melanosomes and potassium superoxide. (**A**) Kinetics of accumulation of fluorescent decay products. The incubation medium contained 0.1 M K–phosphate buffer, pH 7.4, 1.5 × 10^8^ melanosome granules/mL, and 70 mM KO_2_. Curves 1–5—incubation time: 0, 30, 60, 90, and 180 min, respectively. Excitation wavelength—450 nm; (**B**) concentration of TBA-reactive products in water-soluble fractions obtained by oxidation of melanosomes with potassium superoxide. Column 1—original, non-oxidized melanosomes, 2–15 mg KO_2_ added, 3–30 mg KO_2_ added; Data are means (means ± SD) of four assays; (**C**) comparative histogram of carbonyl ion signals in samples of non-oxidized (grey bars) and oxidized KO_2_ (blue bars) RPE melanosomes; an average of 14 experiments, * denotes Student *t* test, *p* value < 0.001; (**D**) kinetics of superoxide formation upon irradiation with blue light for the products of oxidative degradation of RPE melanosomes. The concentration of superoxide dismutase added was 10 µg/mL. Data are means (means ± SD) of three assays.

**Figure 4 ijms-24-13099-f004:**
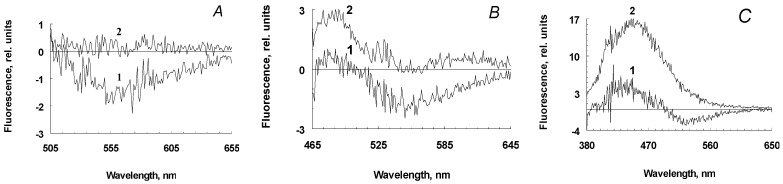
Differential fluorescence spectra of water-soluble products obtained from the RPE pigment granules irradiated and not irradiated with blue light. Excitation wavelength: (**A**)—488 nm, (**B**)—450 nm, and (**C**)—365 nm. Curves 1 and 2 are the fluorescence spectra of water-soluble products obtained by irradiation with melanosome and melanolipofuscin granules, respectively.

**Figure 5 ijms-24-13099-f005:**
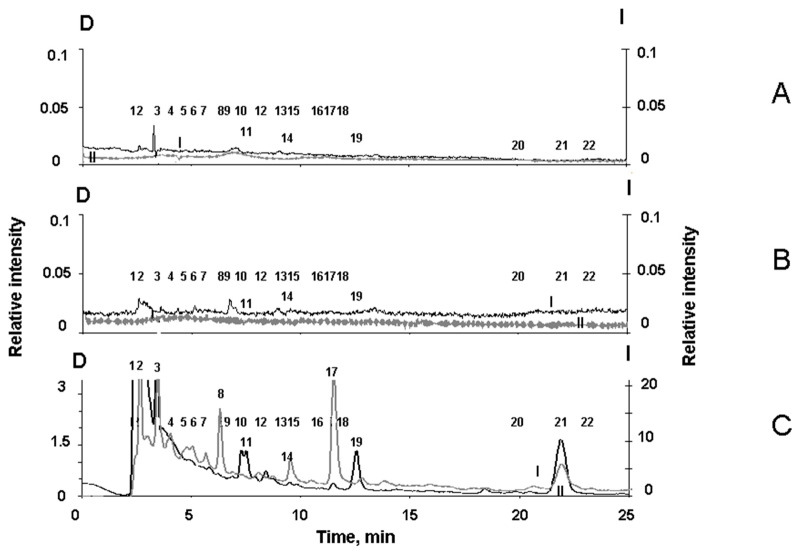
Comparative chromatograms for water-soluble fractions from RPE melanosomes irradiated by light at different intensities. Water-soluble fractions were obtained from (**A**) initial (dark) melanosomes; (**B**) melanosomes irradiated by blue light from a LED source at an energy of 4 mW/cm^2^ for 1.5 h; (**C**) melanosomes irradiated with visible light from a halogen lamp at an energy of 100 mW/cm^2^ for 2.5 h. *y*-axis, D (left) detection by absorption at 270 nm (curves I); *y*-axis, I (right) fluorescence detection (excitation wavelength: 270 nm, emission wavelength: 380 nm (curves II)). Numerals (1–22) indicate the peaks in the chromatograms. The chromatogram for each kind of sample was obtained in three independent experiments. Peak numbers were assigned according to the washout time from the chromatographic column.

**Figure 6 ijms-24-13099-f006:**
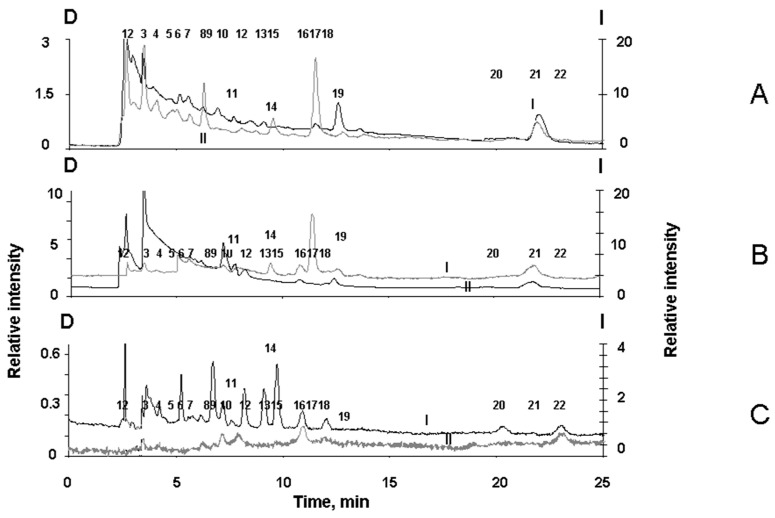
Chromatograms of water-soluble fractions from RPE pigment granules. (**A**) Products of degraded melanosomes oxidized by superoxide; (**B**) and (**C**) are degradation products of melanolipofuscin and lipofuscin granules, respectively, formed upon irradiation of the granules with blue light from an LED lamp. *y*-axis, D (left, curves I)—detection by absorption at 270 nm; *y*-axis, I (right, curves II)—fluorescence detection at an excitation wavelength of 270 nm and emission wavelength of 380 nm. The numbers (1–22) indicate the peaks in the chromatograms. The chromatogram for each kind of sample was obtained in three independent experiments.

**Figure 7 ijms-24-13099-f007:**
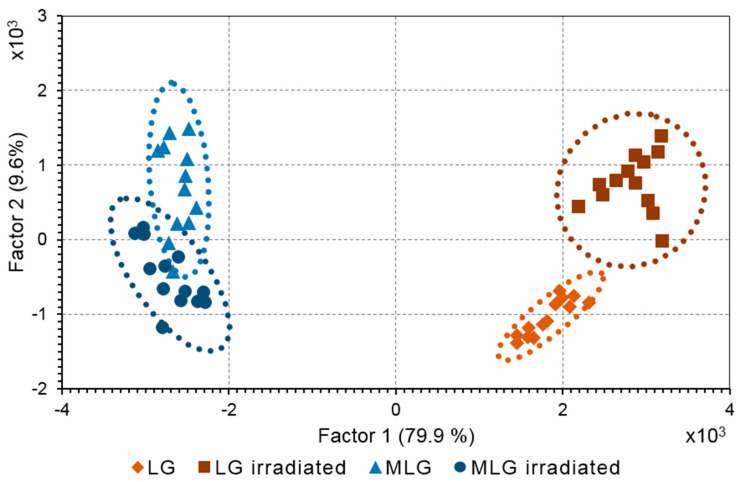
PCA score plot of RPE pigment granules. Suspension of lipofuscin (LG) and melanolipofuscin (MLG) granules were irradiated by a white light LED source at an energy of 20 mW/cm^2^ for 2.5 h. Ellipse indicates a 95% confidence interval. Each symbol represents a single experiment.

**Figure 8 ijms-24-13099-f008:**
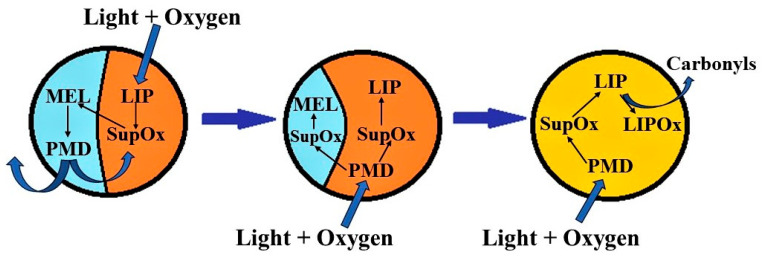
Scheme of the mechanisms involved in melanin degradation in the melanolipofuscin granule. Abbreviations: MEL—melanin; LIP—lipofuscin; PMD—melanin degradation products; LIPOx—oxidized lipofuscin; SupOx—superoxide radicals.

**Table 1 ijms-24-13099-t001:** The ions with enlarged yields in melanolipofuscin granules according to mass spectra data.

Ions	*m*/*z*	Ions	*m*/*z*	Ions	*m*/*z*
CH_2_N^+^	28.02	C_3_H_3_O^+^	55.02	C_2_H_6_N_3_^+^	72.05
C_2_H_4_^+^	28.03	C_3_H_6_N^+^	56.05	C_2_H_7_N_3_^+^	73.06
CH_4_N^+^	30.04	CH_6_N_3_^+^	60.06	C_4_H_6_NO^+^	84.04
C_2_H_3_O^+^	43.02	C_4_H_6_N^+^	68.05	C_5_H_10_N^+^	84.08
CH_2_NO^+^	44.01	C_4_H_5_O^+^	69.03	C_4_H_5_O_2_^+^	85.04
C_2_H_6_N^+^	44.05	C_5_H_10_^+^	70.07		

## Data Availability

The data presented in this study are available in the article and on request from the corresponding author.

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
