# Peer review of "Understanding the Mechanism of Light-Induced Age-Related Decrease in Melanin Concentration in Retinal Pigment Epithelium Cells"

_ijms, 2023, doi:10.3390/ijms241713099_

Round 1

Reviewer 1 Report

The manuscript presented by  Dontsov et al, aims to define the mechanisms of melanosome degradation. The team has a long track record in this field, and the mechanism suggested in this paper and summarized in Figure 8 is  interesting and supported by solid data.  However, some improvements are needed to reach optimal quality:

1.It needs to be clear what the ratio of melanolipofuscin  to lipofuscin granules is. The described mechanism would not make much sense if the amount of melanolipofuscin granules were very small because it would have no physiological relevance, nor would the suggested mechanism be true if this ratio did not change with age in favour of lipofuscin granules. This issue needs proper clarification and since samples have been obtained from human eyes from 40 to 75 years of age, it would be interesting to provide data on age dependent ratio of melanolipofuscin vs lipofuscin granules.

2The beginning of the summary needs a little more background on what melanosomes are and their physiological function.

2.The figures need to improve in quality, it is necessary to homogenize fonts, styles and scales of the graphs within a figure. Figure 3 C and D and figure 8 are blurred. Poorly elaborated figures can detract from  work credibility.

3.Clarify what rel.units means, relative units or arbitrary units?.

4.In figure 2 apparently oxidized and non-oxidized forms are represented, but seems that is not  possible. Figure legends should be self explanatory and containing information regarding only the corresponding figure.

The quality of English is generally good, and text is coherent.

Reviewer 2 Report

The manuscript contains new interesting results of the study of photooxidation processes in melanin-containing structures leading to photodegradation of the retina. The results were obtained using a complex of modern methods of biochemistry, chromatography and mass spectrometry. A scheme of the mechanism of melanin and lipofuscin degradation in melanolipofuscin granules is proposed. The work does not induce serious remarks, however, it contains a number of inaccuracies and unsuccessful editorial moments.

1. The legend to Figure 1 is absent.

2. Lines 100-101 - obviously here should be indication of curve 1, not 2.

3. Lines 152-154. This description from the Methods on lines 366-369 and should be omitted.

4. Figure 4. The caption is too detailed. The results of measurements of the magnitude of the fluorescence response at different excitation wavelengths should be transferred to Results. Please, clarifiy the following:

What is the average value of the emission intensity and how was it determined?

If we mean the magnitude of the wavelengths at which the maximum response is observed, then it does not correspond to those indicated in the caption: instead of 535, 525 and 460 nm, they are rather at 555, 535 and 435 nm.

What do the dotted straight lines in this figure represent?

5.Designations at Figure 5 are ill-sorted. The letter designations of axes and panels (A,B….I) using the same font make it very difficult to understand the figure.

In addition, the curves in the figure are designated as latin I and II, and in the caption as arabic 1 and 2; also the designation of panel B is given as b.

6. Lines 351-353. Please indicate (or give reference) why were such wavelengths of 365 nm, 450 nm, 470 nm, and 488 nm chosen for by measuring the emission intensity?

7. The Methods do not indicate the use of statistics.

Reviewer 3 Report

In their study, the authors describe the degradation of melanin in RPE cells in the presence of lipofuscin and blue light. The manuscript is of high interest and generally well written and performed. Some aspects need to be addressed before publication can be considered

Major concern

It is not clear how many independent experiments were conducted. The authors state that 40 human eyes of different ages were used, but it is not clear whether this translates to 40 different experiments, or if the eyes were pooled, etc. Also, no statement concerning statistics if given, this needs to be added.

Figures

The authors need to revise the figure legends. For all figures, please include the number of experiments conducted.

Figure 1 is not depicted as such and the figure legend is missing.

Figure 3, a statement about the number (n) of experiments and statistics should be given in the legend. Furthermore, the legend seems to be part of the main text body.

Figure 4, the legend has a different font.

Figure 5, this is quite confusing. The designation of the different diagrams are A, B, C on the right hand side, while all diagrams have a “D” on their left hand sight, which looks light a designation but is not. I am not sure what the D stands for. Furthermore, in the legend, the designations are A, (b), and C. Please harmonize the designations.

Figure 8, it is not clear what part of the text belongs to the legend

Other

Please explain all abbreviations at first mention (e.g. TBA, line 164)

Line 167 – 168 “This is quite a lot…”, I do not understand what the authors are trying to express

Line 236 and 239, it is not clear why the peaks are given these numbers

Please leave a blank line between paragraph and new caption

Please leave a blank line between text and table

Line 279, “However, it was unclear did the fragments originate…”, please rephrase

Line 297, ff, “Previously, we found that the concentration of melanin […] does not change with age”. The authors used RPE cells of donors of different ages, did the authors find any difference in the behavior of these different aged melanosomes or were these pooled, irrespective of the age of the donor?

Line 345 “in some experiments”, again, it is not clear how many experiments were condcuted

Only minor editing of English language required
